# Use of Filters to Smooth Out Signals Collected through Mobile Devices in the Static and Dynamic Balance Assessment: A Systematic Review

**Karina Sá** [1], **Givago Souza** [2], **Bianca Callegari** [3], **Anderson Belgamo** [4], **André Cabral** [5], **José Gorla** [1] and **Anselmo Costa e Silva** [6,*]

1   Adapted Physical Activity and Sport Laboratory (LAFEA), State University of Campinas, Campinas 13083-851, Brazil; k219015@dac.unicamp.br (K.S.); gorla@fef.unicamp.br (J.G.)
2   Tropical Medicine Nucleus, Federal University of Pará, Belém 66050-160, Brazil; givagosouza@ufpa.br
3   Human Motricity Studies Laboratory (LEMOH), Federal University of Pará, Belém 66050-160, Brazil; callegari@ufpa.br
4   Department of Computing, Federal Institute of Piracicaba, Piracicaba 13414-155, Brazil; anderson@ifsp.edu.br
5   Faculty of Physiotherapy, State University of Pará, Belém 66050-540, Brazil; andre.cabral@uepa.br
6   Adapted Physical Activity Laboratory (LAFA), Federal University of Pará, Belém 66075-110, Brazil
*   Correspondence: anselmocs@ufpa.br

**Abstract:** Background: When performing motion analysis using sensors, the signal often comes with noise and it is necessary to use filters to exclude unwanted frequencies. For this reason, the objective of this work was to carry out a systematic review on the filters used in data recorded from smartphone applications for static and dynamic balance assessment. Methods: A systematic literature review was performed on the PubMed, ScienceDirect, Scopus, Technology Research and Web of Science databases, using the search strategy: smartphone, "mobile technology", evaluation, "postural stability", and balance. Results: 427 articles were found (PubMed = 107; ScienceDirect = 67; Scopus = 106; Web of Science = 95; Technology research database = 52). After applying the inclusion criteria and removing duplicates, nine studies were eligible for this review. In these studies, the fourth-order Butterworth low-pass filter was the most applied (N = 6) and the cutoff frequency of 4 Hz (N = 2) was the most frequent. Conclusions: In general, few studies have adequately described the filter used in signal processing. This step, when hidden, negatively affects the reproducibility of studies. Understanding and describing the signal processing is important not only for the correct description of the results but also for the reproducibility of the studies.

**Keywords:** balance; smartphone technology; signal processing; filters; movement analysis

## 1. Introduction

Static and dynamic balance involves complex processes integrated by the central nervous system (CNS) through vestibular, visual, and proprioceptive systems, which coordinate adjustments in order to maintain control of the center of mass (COM) within the base of support during static [1] or dynamic balance [2]. These mechanisms can be affected by changes in the visual, vestibular, or proprioceptive systems [3], such as those that occur in the aging process in conditions such as sarcopenia and osteoporosis [4], which contribute to a loss of body support. Currently, there is an interest in methods to quantify balance quantitatively in clinical settings. Therefore, microelectromechanical systems (MEMS) of cell phones have been employed. MEMS are miniature sensors, such as accelerometers and gyroscopes, that obtain inertial variation data, allowing the registration of body movement during static balance or motor execution. They are embedded in modern smartphones, allowing users to access registered data using existing applications [5]. Such applications have been used in papers on balance control [6], gait changes detection in people with Parkinson's disease [7], and the assessment of risk of falls in the elderly [8].

During the data acquisition, interferences may occur generating signal noise (i.e., the vibration generated by a car when passing near the experiment location, even the subject's respiratory chest movement, among others). Noises are unwanted signals that interfere with or distort the desired signal, which can generate erroneous readings, making the results uncertain and inaccurate [9]. Noise will always be present in any data acquisition; however, it is possible to minimize its presence by using some kind of filter to allow for a more consistent data assessment. Therefore, the filters were created to minimize the influence of noise on the recorded data. Andrei Kolmogorov (1903–1987) and Norbert Wiener (1894–1964) created the basis for estimation theories that were later used to develop the theory of data prediction, filtering, and smoothing [10]. Since then, filters have been used to smooth the presence of noise, making data more reliable by reducing the signal present at unwanted frequencies [11].

Figure 1 presents the raw and filtered data of a healthy male volunteer performing a timed up-and-go test (i.e., commonly used to assess the risk of falls in the elderly population). The accelerometry data were recorded using a triaxial sensor (MetaMotionCTM, MBIENTLAB-INC) attached to the volunteer's lumbar region. In this image, the raw data (A), the Butterworth filter (B), and the Kalman filter (C) are displayed.

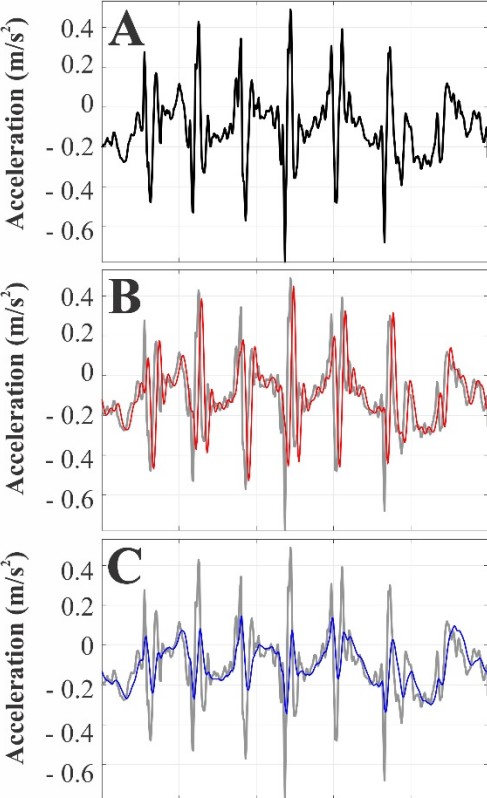

**Figure 1.** Timed up-and-go test inertial recordings: (**A**) raw data (without filters), (**B**) data with the Butterworth filtering process, and (**C**) data with the Kalman filtering process.

These filtering processes are better described in papers using surface electromyography, force platform, and kinematic recordings, and still need to be better described when using mobile devices. Knowing the existing filters and standardizing the signal processing is important because it enables scientific reproducibility. This survey would help researchers and clinicians who are interested in working with the signals from sensors embedded in smartphones, since knowing the signal processing is an important step in obtaining reliable results. The objective of this paper was to perform a systematic review of the filters used in data recorded from smartphone applications for the assessment of static and dynamic balance.

## 2. Materials and Methods

A systematic search was carried out in the PubMed, ScienceDirect, Scopus, Technology Research and Web of Science databases and collected through other sources such as the references of the selected articles themselves, using the search strategy: smartphone, "mobile technology", evaluation, "postural stability" and balance. The search strategy was originated through the PICO strategy, in which P = concerns the studied population (general population for balance assessment), I = is about the research interest (balance assessment using APP on mobile devices), C = represents the comparison (use of filters) and O = is the expected outcome (to define the most used filters and the algorithms used in the filters).

The eligibility of the papers was first based on the title and abstract. Then, articles that met the following inclusion criteria were included: (1) original articles (not review) (2) that used mobile devices (e.g., tablets, smartphones, among others) for the assessment of static and dynamic balance or posture, and (3) that presented, in the methodology, the specifications of the filter used for signal processing. Articles using sensors that are not embedded in mobile devices and/or without a complete description of data filtering were excluded from this review. The systematic review was carried out in accordance with the Preferred Item Reporting Guidelines for Systematic Reviews and Meta-Analysis (PRISMA) [12], described in Figure 2. The search for and eligibility evaluation of the articles were performed independently by two researchers.

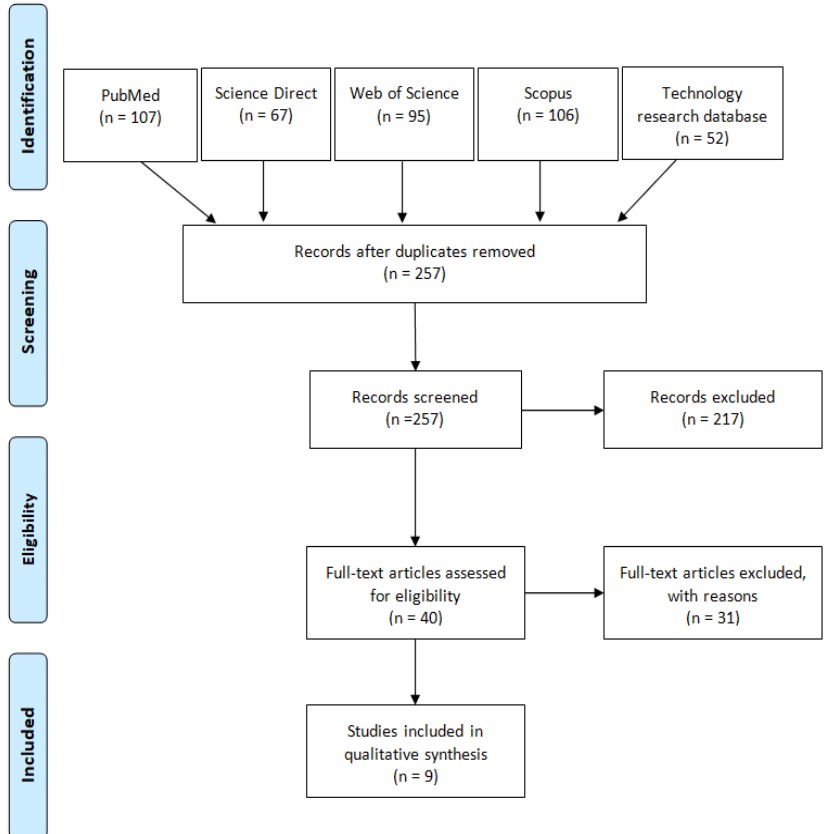

**Figure 2.** PRISMA flow diagram [12].

Data extraction and organization included the author's name and year, objectives, device used in the study, filters used for data processing, and the main results of the study. The quality assessment of the articles was performed using the Evaluation tool for Cross Sectional Studies (AXIS tool) [13]. This tool consists of 20 questions distributed among the introduction, methods, results, discussion, and other aspects, which evaluate the richness of bias. This evaluation is subjective and is performed based on the reading of the article.

Evaluation of the methodological quality of the studies had a guide character, but was not a criterion for the exclusion of the articles in the review.

## 3. Results

### 3.1. Article Selection

Searching the descriptors, a total of 427 articles were identified (PubMed = 107; ScienceDirect = 67; Scopus = 106; Web of Science = 95; Technology Research Database = 52). After applying the inclusion criteria and removing duplicates, nine studies were eligible for this review (Figure 2). These articles are described in Table 1.

**Table 1.** Article Descriptions.

| Author/ Year | Purpose | Device and Application | Filter | Results |
|---|---|---|---|---|
| Yamada et al., 2012 [20] | The authors evaluated a smartphone gait analysis app in patients with rheumatoid arthritis (RA). | -smartphone (size: 63-mm width, 119-mm height, 13.1-mm depth; weight: 139 g; Xperia SO-01B; Android 2.1; Sony Ericsson Mobile Communications Japan, Inc.) -Application: Unnamed. The application developed in the android environment. | Low-pass filter | The RA group showed significantly lower scores for walking speed, correlation peak (AC), and coefficient of variance (CV) than the control group. The peak frequency (PF) (gait cycle) was mildly associated with gait speed ($p < 0.05$). The results suggest that some gait parameters recorded using the smartphone represent an acceptable gait assessment tool in patients with RA. |
| Wai et al., 2014 [19] | Introduce the iBEST (Intelligent Balance Assessment and Stability Training) app to assess balance. | -Smartphone -Application: iBEST (intelligent balance assessment and stability training). | Kalman filter (used to fuse synthetic orientation obtained from the smartphone library and orientation estimates from measurements from physical sensors' measurements). | The feasibility study showed an average accuracy of 90.22% using the smartphone to classify the specified BBS test elements. |
| Ozinga & Alberts 2014 [14] | Check kinematic data collected using a mobile device to characterize postural stability in the elderly. | -iPad and motion analysis system (Motion Analysis Corporation Eagle System; Santa Rosa, CA, USA) with eight infrared Eagle digital cameras. -Application: Cleveland Clinic Balance Assessment App (CCBApp). In-house application. | Fourth-order, low-pass Butterworth filter. | The correlation between the two systems (iPad and motion analysis system) was significant in all balance conditions and outcome measures: peak to peak (r = 0.70–0.99), normalized path length (r = 0.64–0.98), linear acceleration root (r = 0.73–0.99), linear and angular acceleration 95% of the volume (r = 0.96–0.99) of and total power at different frequencies (r = 0.79–0.92). |
| Pan et al., 2015 [22] | Design, develop, and evaluate a cloud-based mobile health mobile application prototype for monitoring the main symptoms of Parkinson's disease at home. | -Smartphone -Application: PD Dr (Parkinson disease Dr) | Low-pass filter | For the detection of hand resting tremor, the sensitivity was 0.77 and the precision was 0.82. For the detection of walking difficulties, the sensitivity was 0.89 and the precision was 0.81. |

**Table 1.** *Cont.*

| Author/ Year | Purpose | Device and Application | Filter | Results |
|---|---|---|---|---|
| Ozinga et al., 2015 [17] | To determine whether the kinematic data measured by hardware inside a tablet device was of sufficient quantity and quality to characterize postural stability in people with Parkinson's. | -iPad (3rd generation Apple iPad, Cupertino) and kinematic system (Motion Analysis Corporation Eagle System; Santa Rosa, CA, USA). -Application: Cleveland Clinic Balance Assessment App (CCBApp). In-house application. | Fourth-order, Butterworth filter, low-pass with 4 Hz cutoff frequency | The motion capture system and tablet provided similar measures of stability between groups. Within the patient population, the correlation between the two systems for peak-to-peak, normalized path length, root mean square, 95% volume, and total power values ranged from 0.66 to 1.00. |
| Alberts et al., 2015 [18] | Determine whether data collected from a consumer electronics device (iPad2) provide sufficient resolution of center of gravity (COG) movements to accurately quantify postural stability in healthy young people. | -Ipad2 and e NeuroCom force plate -Application: Cleveland Clinic Balance Assessment App (CCBApp). In-house application. | Fourth-order, low-pass Butterworth filter with a cutoff frequency of 1.25 Hz. | Limits between the 2 devices ranged from 0.58 to 0.58 in the NeuroCom Sensory Organization Test (SOT) condition 1 and from 2.98 to 1.38 in SOT condition 5. The highest absolute value of the measurement error within the 95% confidence range for all conditions was 2.98. The mean absolute percent error analysis indicated that iPad2 tracked NeuroCom COG with a mean error ranging from 5.87% to 10.42% of the NeuroCom measurement under SOT conditions. |
| Ozinga et al., 2017 [16] | Validate a mobile device platform that characterizes posture stability. | -iPad -Application: Cleveland Clinic Balance Assessment App (CCBApp). In-house application. | Fourth-order Butterworth filter, low pass with 4 Hz cutoff frequency. | The mobile device platform was able to distinguish in all conditions Parkinson's participants from controls. Peak-to-peak balance metric was significantly higher in Parkinson's disease compared to controls ($p < 0.01$ for all tests). |
| Koop et al., 2018 [21] | Determine whether the biomechanical metrics of a mobile inertial measurement device unit were sensitive to characterize the effects of antiparkinsonian medication during the Timed Up and Go (TUG) test. | -iPad (Apple, Inc. Cupertino, CA, USA) -Application: Cleveland Clinic Mobility and Balance Application (CC-MB). In-house application. | -Acceleration = low-pass, 20 Hz zero-lag 4th order Butterworth. -Angular velocity = zero-lag 4th order Butterworth filter with cutoff frequencies of 0.25 Hz and 20 Hz. | The mobile device detected significant improvements associated with antiparkinsonian drugs. The platform provides objective reports immediately after clinical evaluations. |
| Hsieh et al., 2019 [15] | Determine whether an accelerometer built into a smartphone can measure static postural stability and distinguish older adults with high levels of risk of falling. | -Smartphone (Samsung Galaxy S6, Samsung, Seoul, South Korea) and Force Platform (Bertec Inc., Columbus, OH, USA) -Application: Not reported | Fourth-order Butterworth filter and low-pass at a frequency of 10 Hz (force platform) | The accelerometer built into a smartphone had moderate to strong correlations with the force platform during challenging equilibrium conditions ($\rho = 0.42$–$0.81$; $p < 0.01$–$0.05$). |

Most articles used applications to assess the postural stability (N = 5) of different populations: elderly or older adults [14,15], people with Parkinson's disease [16,17] and young

people [18]. Furthermore, studies assessed fall risks (N = 2) [15,19] and gait (N = 2) [20,21]. To verify the utility of applications, the articles compared the data collected from applications with kinematics (N = 3) [14,17,18], the force platform (N = 1) [15], and scales and clinical evaluations (N = 4) [16,19,20,22].

*3.2. Quality Assessment*

The evaluation of methodological criteria was performed using the AXIS tool, determining which articles had good methodological quality. The results of this evaluation can be consulted in the Supplementary Material Table S1.

*3.3. Operational System and Mobile Device Apps*

The most present operating system on mobile devices was IOS from the Apple company (N = 5) [14,16–18,21], followed by the Android operating system (N = 2) [15,20]. There were studies that did not inform about the type of device or its operating system (N = 2) [19,22]. Of the total articles, eight used their own developed applications and one did not report application usage.

*3.4. Filters*

It was observed that the fourth-order Butterworth low-pass filter was reported more frequently (N = 6) [14–18,21], followed by the low-pass filters (without specification) (N = 2) [20,22] and the Kalman filter (N = 1) [19] and the cutoff frequency adopted varied from 1.25 Hz to 20 Hz. The most reported frequency was 4 Hz (N = 2) [16,17], then 20 Hz (N = 1) [21], 1.25 Hz (N = 1) [18], 10 Hz (N = 1) [15], 0.25 Hz (N = 1) [21] and not reported (N = 4) [14,19,20,22]. In general, signal processing occurred after data collection (N = 8), through analysis software [14,15,17,18,20,21] or cloud [19,22]. Only one study had signal processing in the application [16].

## 4. Discussion

The correct choice of the filter type and the frequency adopted is also important for better signal processing. The results in this review showed that the fourth-order Butterworth low-pass filter with a cutoff frequency of 4 Hz was the most used feature employed in the studies.

*4.1. Operational System*

Most of the studies used the IOS system, owned by Apple. Basically, the operating system of a mobile device is the platform responsible for performing the interaction between the device and the subject. Hence, the device's operating system directly interferes in the choice of the application. Concerning the applications used in the studies, five of the nine articles used self-created applications in this development environment [14,16–18,21], performing their validation for their specific purposes. This demonstrates a growing interest in the field of technological development for health. There is a range of applications available on shopping websites for operating systems with several functionalities focused on the health area. In this expansion scenario, studies are important to validate the use of these applications.

*4.2. Use of Filters*

There is a lack of this description in mobile device studies, and perhaps that is why its use depends on the researcher's choice. In this review, the filters used in the studies were Butterworth and Kalman. The Butterworth filter is a type of electronic filter developed to have a flat frequency magnitude response, as far as possible mathematically, in a defined passband. The Butterworth filter has five frequency orders and its use varies according to the desired response frequency [23]. The effect of noise suppression and signal distortion produced by the Butterworth filter can be investigated and controlled regardless of whether it is a linear operator [24]. The Kalman filter can be used to integrate two sources of

information in a fusion of sensors and obtain the best estimate of unknown variables. This is what happens, for example, in the Inertial Measurement Unit (IMU), which is used in recordings with mobile devices through MEMS. In these measurements, two sensors (accelerometer and gyroscope) are fused to estimate an object's motion, speed and orientation throughout accelerations and angular data. The conventional Kalman filter is linear with Gaussian noise and could present problems in modeling nonlinear functions or linear functions with non-Gaussian noise [25]. To improve the applicability of the Kalman filter, scientists have created variants of the calculations, and it is recommended that the modeling of IMU error sources must be performed separately. Through the studies reviewed, the choice of filters used and the choice of the cutoff frequency are not standardized in the literature, since studies with similar objectives have chosen different frequencies and filters. However, while reading the articles, we identified that in studies focused on static balance analyses or specific movements, the cutoff frequencies were lower. On the other hand, in studies about gait or dynamic balance, in which several movements are occurring at the same time, the cutoff frequencies were higher. This may be related to the fact that in dynamic activities the sensor suffers more from noise interference. Therefore, describing the signal filtering system is important for replicating the experiment and standardizing data processing for balance analysis, using mobile devices.

The studies in this review do not properly describe signal processing, but the majority stated that the data had a good correlation and validation with the gold standard equipment, such as force platform and kinematics. Even so, when authors do not describe the filtering process, the results can be questioned, and the work methodology becomes fragile, interfering with its replication. The literature shows that the impossibility of reproducing some studies has reached high values in the scientific community, and this demonstrates that the published results are less reliable than has been claimed [26]. Many causes contribute to this phenomenon, and we highlight the analysis and reporting of data. Data analysis and reporting needs to be written in clear detail to achieve the principle of scientific reproducibility.

### 4.3. Authors' Suggestions for the Replicability of Articles

The authors believe that to strengthen the reproducibility of studies that use signals collected through mobile devices, it is necessary to report some items in the study methodology. Such items are filter name, cutoff frequencies, sample rate of the record, and whether the signal processing will be performed in an external way or within the application.

### 4.4. Limitations and Future Directions

One possible limitation of this study is that it only considered MEMS/IMU on mobile devices, and we did not address this technology present in wearable sensors such as smartwatches and commercial IMUs. In this sense, it is worth mentioning that the MEMS/IMU signal can suffer interference from several factors, as discussed earlier. With that in mind, when considering other technologies and comparing with mobile devices, the weight of the device and the sampling rate, as well as the device positioning, may differ from one device to another and so the signal may change. Therefore, carrying out a study comparing the differences between devices and verifying how this interferes with the signal would be interesting to enrich the discussions on the subject. In addition, based on the results found here, we believe that carrying out studies that compare signals using different filters and different cutoff frequencies can be the first step towards a standardization of the analysis of signals collected through sensors.

### 5. Conclusions

The fourth-order Butterworth low-pass filter was the most used filter in the studies present in this review. However, the use of filters to process data recorded using a mobile device still needs further study, and apparently no filter is yet considered the best to be

used. Understanding and describing signal processing is important not only for the correct description of the results but also for the reproducibility of the studies.

**Supplementary Materials:** The following supporting information can be downloaded at: https://www.mdpi.com/article/10.3390/app12136579/s1, Table S1: Appraisal tool for Cross-Sectional Studies (AXIS).

**Author Contributions:** Conceptualization, K.S. and A.C.e.S.; methodology, K.S.; writing—original draft preparation, K.S., B.C., A.B. and A.C.; writing—review and editing, G.S., J.G. and A.C.e.S. All authors have read and agreed to the published version of the manuscript.

**Funding:** This work was supported by research grants from the Coordenação de Aperfeiçoamento de Pessoal de Nível Superior—Brazil (CAPES), finance code 001, Brazilian funding agencies CNPQ-IC UFPA, PAPQ-PROPESP-UFPA, Amazon Paraense Foundation of Studies (FAPESPA, No. 2019/589349), and the Research Funding and the National Council of Research Development (CNPq/Brazil, No. 431748/2016-0). G.S. was a CNPq Productivity Fellow (No. 310845/2018-1). The funders had no role in the study design.

**Institutional Review Board Statement:** Not applicable.

**Informed Consent Statement:** Not applicable.

**Conflicts of Interest:** The authors declare no conflict of interest.

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
