# Peer review of "Use of Filters to Smooth Out Signals Collected through Mobile Devices in the Static and Dynamic Balance Assessment: A Systematic Review"

_applsci, doi:10.3390/app12136579_

Round 1
Reviewer 1 Report
Dear corresponding Author thanks for submitting. I appreciated your work.
Author Response
Dear reviewer, thank you for the positive feedback.
Reviewer 2 Report
A systematic review about the filters used to smooth signals of MEMS/IMU when assessing static and dynamic balance is of interest as well as clearly contributes to studies reproducibility, central in scientific studies.
However, in my opinion the search strategy and the review focus on mobile devices (phones, tablets), that is too specific, when it should focus on the technology used on mobile devices, MEMS/IMU, and probably it would have included more studies and allowed a powerful review. The mobile devices had MEMS or only IMU, therefore if the goal is to analyze filters acceleration data why not use a search strategy for mobile IMU, independently if used on a mobile phone/tablet or another device.
Suggest presenting and discuss if different filters are used to smooth data considering if static or dynamic balance are being analyzed.
Minor aspects:
Title – In my opinion should mention that is a systematic review
Filliation - 1st filiation present the Lab with capital letters
L39 – improve sentence readability.
L105-108 – In my opinion is not appropriate to begin the results section with a definition of what is expected in theory to be presented on the section of a systematic review.
L125 – confirm “without”.
Author Response
Dear reviewer,
Thanks for the suggestions. We take them all into consideration and hope to have taken care of them to the fullest.

Reviewer 3 Report
The objective of this review is to examine the usage of filters to smooth out static and dynamic balance assessment signals gathered via mobile devices. This is a significant article on the subject. However, the author need to address many concerns.
1. Kindly clarify how did you assess methodological quality of each included study?
2. Authors are required to discuss not only the qualitative aspects of their work but also their findings.
3. Authors shall acknowledge potential limitations of their study.
4. Authors shall include a few figures to improve understanding of review findings.
5. Authors have only enumerate the different filtering techniques used in the included study. Authors shall present in-depth analysis about different kind of filtering techniques used in the included studies.
6. Authors shall provide a few research direction based on this review for future studies.
Author Response

(The authors gave the same response as above.)

Round 2
Reviewer 2 Report
I consider that the authors have improved the paper considering reviewers comments. I maintain my opinion that the fact that the review only focusses on MEMS/IMU on mobile devices, and not on the technology. In authors answering they wrote that "In this sense, many studies have been carried out using mobile devices", however when we read the systematic review 9 studies seems low. In my opinion this topic could be analysed considering not only mobile devices, but MEMS/IMU used for the analysis of static and dynamic balance assessment or introduce on the paper the explanation of the difference in MEMS/IMU signal on a mobile device, compared with the same technology (MEMS/IMU) but not incorporated on a mobile device, not allowing to consider other studies.
Author Response
Dear reviewer, we took your suggestion into consideration and thus added some points to the manuscript.

Reviewer 3 Report
Authors have made the required changes in the revised manuscript.
Author Response
Thank you for your suggestions.